# SERPINF1 Mediates Tumor Progression and Stemness in Glioma

**DOI:** 10.3390/genes14030580

**Published:** 2023-02-25

**Authors:** Lairong Song, Xulei Huo, Xiaojie Li, Xiaoying Xu, Yi Zheng, Da Li, Junting Zhang, Ke Wang, Liang Wang, Zhen Wu

**Affiliations:** 1Department of Neurosurgery, Beijing Tiantan Hospital, Capital Medical University, Beijing 100071, China; 2China National Clinical Research Center for Neurological Diseases, Beijing 100071, China

**Keywords:** SERPINF1, glioma, progression, stemness, Notch signaling pathway

## Abstract

Serpin family F member 1 (SERPINF1) reportedly plays multiple roles in various tumors; however, its clinical significance and molecular functions in glioma have been largely understudied. In the present study, we analyzed the prognostic value of SERPINF1 in three independent glioma datasets. Next, we explored the molecular functions and transcriptional regulation of SERPINF1 at the single-cell level. Moreover, in vitro experiments were conducted to evaluate the roles of SERPINF1 in the proliferation, invasion, migration, and stemness of glioma cells. Our results showed that a higher expression of SERPINF1 correlated with a poor overall survival rate in glioma patients (hazard ratio: 4.061 in TCGA, 2.017 in CGGA, and 1.675 in GSE16011, *p* < 0.001). Besides, SERPINF1 knockdown could suppress the proliferation, invasion, and migration of glioma cells in vitro. In addition, SERPINF1 expression was significantly upregulated in glioma stem cells (GSCs) compared to parental glioma cells. Knocking down SERPINF1 impaired the sphere formation of GSC-A172 and GSC-LN18. Bioinformatics analysis revealed that Notch signaling activation was closely associated with high SERPINF1 expression at the single-cell level. Furthermore, STAT1, CREM, and NR2F2 may participate in the transcriptional regulation of SERPINF1 in glioma. Overall, our results suggest that SERPINF1 may be a candidate prognostic predictor and potential therapeutic target for glioma.

## 1. Introduction

Glioma is the most frequent primary malignant tumor in the central nervous system [1]. Due to the infiltrative nature of glioma, complete tumor resection is often challenging [2]. Maximal surgical resection followed by postoperative radiotherapy and chemotherapy remains the standard of treatment for glioma [3]. Nonetheless, rapid recurrence is common, especially in glioblastoma (GBM) patients. It has been reported that the five-year survival rate of GBM patients is approximately 5.8% [4]. Due to the resistance of glioma to standard radiotherapy and chemotherapy, new strategies have been attempted, such as immunotherapy, electric field therapy, and oncolytic virus therapy [5]. However, the therapeutic effect remains far from satisfactory [6,7,8]. A better comprehension of the molecular mechanisms underlying tumor progression and therapy resistance is urgently needed.

Serpin family F member 1 (SERPINF1), also known as pigment epithelium-derived factor (PEDF), is a multifunctional secreted protein [9] that can exert an anti-tumor effect by inhibiting angiogenesis [10,11]. Interestingly, it has also been documented to promote the proliferation and invasion of tumor cells [12,13]. In this respect, Li et al. reported that SERPINF1 exhibits contrasting intracellular and extracellular functions in hepatocellular carcinoma [14]. It is highly conceivable that SERPINF1 mediates the balance between the activation of tumor growth and the inhibition of tumor angiogenesis [14]. However, the functional role of SERPINF1 in glioma has not been elucidated in detail.

Glioma stem cells (GSCs) are a subpopulation of glioma cells that are believed to be responsible for the initiation, maintenance, and progression of gliomas [15]. They are characterized by their ability to self-renew and differentiate into multiple cell types, and are thought to be resistant to conventional anti-tumor treatments [15]. GSCs are considered the source of recurrence of gliomas and are therefore a major focus of glioma research. Previous studies have shown that neural stem cells (NSCs) and GSCs are closely related [16,17]. It has been hypothesized that NSCs are the precursors of GSCs [16,18]. Interestingly, SERPINF1 has been found to promote the self-renewal and multi-differentiation potential of NSCs [19]. Therefore, the association between SERPINF1 and GSCs was investigated in our research.

In this study, we assessed the role of SERPINF1 in the clinical stratification of glioma patients in three independent datasets. Additionally, we analyzed the relationship between SERPINF1 expression and tumor stemness in glioma. Furthermore, SERPINF1-related pathways were investigated using single-cell RNA-sequencing (scRNA-seq) data from the Tumor Immune Single-cell Hub (TISCH) database. Finally, the effects of SERPINF1 knockdown on glioma cells and GSCs were evaluated in vitro.

## 2. Materials and Methods

### 2.1. Patients and Datasets

RNA-sequencing (RNA-seq) data and clinical information of 666 gliomas from The Cancer Genome Atlas (TCGA) and 897 gliomas from the Chinese Glioma Genome Atlas (CGGA) were included in this analysis. In addition, the microarray data and clinical information of 248 gliomas were downloaded from the Gene Expression Omnibus (GEO) database (ID: GSE16011). Patients with an overall survival of fewer than 30 days were excluded from the survival analysis. Kaplan–Meier survival analysis was used to evaluate the stratification ability of SERPINF1. Time-dependent Receiver Operating Characteristic (ROC) curves were applied to assess the prognostic value.

### 2.2. Calculation of DNA Methylation-Based Stemness Index

Acquisition of stem-cell-like features is closely related to tumor malignancy [20]. To calculate the stemness index of tumors, Malta et al. built a computational predictive model based on the 450 K DNA methylation data [21]. The workflow is available at https://bioinformaticsfmrp.github.io/PanCanStem_Web/ (accessed on 25 January 2022). The stemness indices of 559 gliomas with 450 K DNA methylation data were calculated and normalized to the range.

### 2.3. scRNA-seq Data Processing of Glioma Samples

The scRNA-seq data of three glioma samples were obtained from the TISCH database (ID: Glioma_GSE131928). The detailed processing flow was described in our previous study [22]. Briefly, a total of 5311 cells were analyzed using the R package “Seurat”. Cells were clustered using FindNeighbors and FindClusters functions with a resolution of 0.6. The t-distributed Stochastic Neighbor Embedding (tSNE) algorithm was used to reduce dimensionality. Next, the FindAllMarkers function was applied to analyze the marker genes of each cell cluster. Cell types were identified based on the marker genes from the CancerSCEM database (https://ngdc.cncb.ac.cn/cancerscem/index (accessed on 1 September 2022)) and the CellMarker database (http://xteam.xbio.top/CellMarker/ (accessed on 1 September 2022)). The R package “clusterProfiler” was applied to perform the KEGG analysis.

### 2.4. Single-Cell Regulatory Network Inference and Clustering (SCENIC) Analysis

To explore the transcriptional regulation of SERPINF1 in glioma cells, we performed SCENIC analysis at the single-cell level [23]. The R package “SCENIC” was applied to infer the potential transcription factors (TFs) of SERPINF1. RNA-seq data of gliomas from TCGA and CGGA were used to verify the correlation between the expression of TFs and SERPINF1.

### 2.5. Cell Culture

Human glioma cell lines A172, U118, SW1088, and LN18 were cultured in Dulbecco’s modified Eagle medium (DMEM; Gibco, NY, USA) supplemented with 10% fetal bovine serum (FBS; BI, Kibbutz, Israel). GSCs were cultured in DMEM/F-12 supplemented with 1 × B27 (Invitrogen, CA, USA), 1×N2 (Invitrogen), epidermal growth factor (20 ng/mL, Abcam, UK), and basic fibroblast growth factor (20 ng/mL, Abcam). All cells were incubated at 37 °C with 5% CO2.

### 2.6. Cell Transfection

For transient SERPINF1 silencing, 2 × 10^5^ glioma cells were seeded into 6-well plates and incubated overnight. Subsequently, glioma cells were transfected with siRNA at a concentration of 50 nM using lipo3000 (Invitrogen). si-SERPINF1-1, si-SERPINF1-2, and negative control siRNA (si-NC) were purchased from Ribio Company (Guangzhou, China). The sequences of siRNAs are presented in Appendix A.

### 2.7. Real-Time Quantitative PCR (RT-qPCR)

Successful silencing of SERPINF1 was verified via RT-qPCR. The detailed protocol was described in our previous study [22]. The relative SERPINF1 expression was evaluated using the 2^−(ΔΔCt)^ method [24]. All quantifications were normalized to GAPDH as an endogenous control. The primer sequences are also detailed in Appendix A.

### 2.8. Western Blot

Western blot was also conducted to verify the silencing efficiency of SERPINF1 in glioma cells. Total proteins of the A172 and LN18 cells were extracted using a Total Protein Extraction Kit (Solarbio, Beijing, China) at 4 °C and then quantified using a BCA Protein Assay Kit (Beyotime, Nantong, China). Next, equal amounts of proteins were separated on 15% sodium dodecyl sulfate-polyacrylamide gel electrophoresis (SDS-PAGE) and transferred to a polyvinylidene fluoride (PVDF) membrane (Millipore, Darmstadt, Germany). After blocking with 5% skim milk for 1 h, the membrane was incubated with a primary antibody against SERPINF1 (1:1000, Abclonal, Wuhan, China) at 4 °C overnight. Subsequently, a secondary antibody (1:10,000, Proteintech, Wuhan, China) was incubated for 1 h at room temperature. The bands were detected using the Chemiluminescent HRP Substrate (Millipore, MA, USA). For the internal reference detection, the same membrane was stripped and then incubated with the primary antibody against β-actin (1:10,000, Proteintech, Wuhan, China) for 1 h at room temperature. The following steps were consistent with those of SERPINF1 detection.

### 2.9. Cell Proliferation Assay

After transfection for 48 h, A172 and LN18 cells were digested and inoculated into 96-well plates at a density of 2 × 10^3^ cells per well. Then, we evaluated cell proliferation using the Cell Counting Kit-8 (CCK-8, Dojindo, Kumamoto, Japan) at 0, 24, 48, 72, 96, and 120 h, respectively. In detail, the cells were incubated in 100 µL of serum-free medium containing 10 µL of CCK-8 solution for 1 h at 37 °C. Subsequently, the absorbance value was detected at 450 nm with a microplate reader (Tecan Spark, Männedorf, Switzerland).

### 2.10. Cell Invasion Assay

A cell invasion assay was performed using Transwell invasion chambers (Corning, NY, USA). The detailed protocol was described in our previous study [22]. After transfection for 48 h, A172 and LN18 cells were digested and resuspended in 200μL of serum-free DMEM at a density of 5 × 10^5^ cells/mL and inoculated into the upper chamber. The invaded cells were detected via crystal violet staining after incubation for 24 h.

### 2.11. Cell Migration Assay

Cell migration capacity was evaluated via a wound healing assay using the 2-well culture insert (Ibidi, Gräfelfing, Germany). After transfection for 48 h, A172 and LN18 cells were digested and inoculated into the culture chamber (4 × 10^4^ cells in 70 μL of complete medium). After overnight incubation, the culture insert was removed, and the complete medium was replaced with a low serum medium (2% FBS). The images of wound healing were taken under a microscope at 0 and 24 h.

### 2.12. Statistical Analysis

All statistical analyses were conducted using R software version 4.1.3. A *p*-value < 0.05 was considered statistically significant.

## 3. Results

### 3.1. High Expression of SERPINF1 Is Correlated with Risk Factors of Glioma

Differential expression analysis of SERPINF1 was performed in 666 gliomas from TCGA and 897 gliomas from CGGA. Transcript per million (TPM) was used as the unit of the gene expression level. Compared to lower-grade gliomas (LGGs), glioblastomas (GBMs) expressed higher levels of SERPINF1 (Figure 1). In addition, SERPINF1 was highly expressed in LGGs with wild-type IDH and 1p19q non-codeletion (Figure 1). These findings substantiated that increased expression of SERPINF1 was associated with a high malignancy in glioma.

### 3.2. High SERPINF1 Expression Predicts Poor Prognosis in Glioma Patients

To evaluate the prognostic stratification capacity of SERPINF1, a Kaplan–Meier survival analysis was performed in three different datasets. Glioma samples were stratified into high- and low-SERPINF1 groups based on the median expression level of SERPINF1. The log-rank test was applied to evaluate the significance of the survival difference between the two groups. Kaplan–Meier survival curves showed that the overall survival of patients in the high-SERPINF1 group was worse than that of patients in the low-SERPINF1 group (Figure 2A–C). Additionally, ROC curve analysis demonstrated that SERPINF1 expression could effectively predict the 3-, 5-, and 8-year survival of glioma patients in all three datasets (Figure 2D–F). Taken together, these findings suggested that increased expression of SERPINF1 predicted poor prognosis in glioma patients.

### 3.3. SERPINF1 Knockdown Inhibits the Proliferation, Invasion, and Migration of Glioma Cells

To evaluate the effect of SERPINF1 on glioma cells, we knocked down the expression of SERPINF1 with siRNA in glioma cell lines A172 and LN18. A RT-qPCR and Western blot were applied to verify the knockdown efficiency of SERPINF1 (Figure 3A,B and Appendix A). The CCK-8 assay revealed that SERPINF1 knockdown significantly inhibited the proliferation of A172 and LN18 cells (Figure 3C,D). Additionally, the Transwell and migration assays showed that the invasion and migration capability of A172 and LN18 cells was decreased following SERPINF1 knockdown (Figure 3E–G). The cellular morphological images of A172 and LN18 after SERPINF1 knockdown are presented in Appendix A.

### 3.4. SERPINF1 Expression Is Closely Related to Glioma Stemness

It is now understood that cancer stem cells play an essential role in the recurrence and therapy resistance of tumors [25]. SERPINF1 has been substantiated to promote the self-renewal and multi-differentiation potential of neural stem cells [19]. Therefore, the relationship between SERPINF1 expression and glioma stemness was investigated. Based on the computational model developed by Malta et al. [21], we calculated the stemness indices of 559 gliomas using 450 K DNA methylation data. The Kaplan–Meier survival analysis verified that glioma patients with a high stemness index experienced worse survival outcomes than those with a low stemness index (Figure 4A). Spearman’s correlation analysis showed that the expression of SERPINF1 was positively correlated with the stemness index in glioma (Figure 4B). The stemness index of gliomas in the high-SERPINF1 group was significantly higher than that of gliomas in the low-SERPINF1 group (Figure 4C).

Furthermore, the relationship between SERPINF1 expression and glioma stemness was explored in vitro. The RT-qPCR revealed that SERPINF1 expression was significantly upregulated when glioma cells were induced into GSCs (Figure 5A). Moreover, knocking down SERPINF1 impaired the sphere formation of GSC-A172 and GSC-LN18 (Figure 5B).

### 3.5. Single-Cell Analysis Reveals Notch Signaling Pathway Is Activated in Glioma Cells with High SERPINF1 Expression

The characteristics of the single-cell dataset have been described in our previous study [22]. A total of 5311 cells from three glioma samples were classified into four major clusters, including glioma cells, oligodendrocytes, myeloid cells, and T cells [22]. Subsequently, 2751 glioma cells were selected for further analysis. Glioma cells were divided into nine subclusters based on the tSNE algorithm (Figure 6A). The feature plot showed that SERPINF1 was significantly highly expressed in the glioma cells of cluster one (Figure 6B). To explore the signaling pathways associated with SERPINF1 expression, we classified the glioma cells of cluster one as the high-SERPINF1 cluster and the rest as the low-SERPINF1 cluster. Next, gene set variation analysis (GSVA) was applied to evaluate the pathway activity score of each glioma cell. A comparison of the high-SERPINF1 versus the low-SERPINF1 cluster showed that cancer-promoting signaling pathways were significantly enriched, including for Notch signaling, E2F targets, mitotic spindle, G2M checkpoint, wnt/β-catenin signaling, and MYC target pathways (Figure 6C). In addition, KEGG analysis at the single-cell level also demonstrated that cancer-promoting signaling pathways were enriched in the high-SERPINF1 cluster (Figure 6D). Interestingly, both GSVA and KEGG indicated that Notch signaling activation was closely related to high SERPINF1 expression at the single-cell level.

### 3.6. SCENIC Analysis Uncovers the Potential TFs Regulating SERPINF1 Expression

We performed the SCENIC analysis at the single-cell level to explore the transcriptional regulation of SERPINF1 in glioma cells. The predicted TFs that may regulate the expression of SERPINF1 are presented in Figure 7A. Next, we calculated the correlations between the expression of the predicted TFs and SERPINF1 using the RNA-seq data of gliomas from TCGA and CGGA (Figure 7B,C). The top five TFs with the highest correlations in both datasets consisted of STAT1, MEOX2, CREM, NR2F2, and IRF3, indicating that they were likely to regulate the expression of SERPINF1 in glioma. Furthermore, we searched the ChIP-sequencing data of the five TFs in the Cistrome Data Browser [26] and found that STAT1, CREM, and NR2F2 were highly enriched in the promoter region of SERPINF1 (Appendix A), which confirmed the potential transcriptional regulatory effect induced by STAT1, CREM, and NR2F2 on SERPINF1. The correlations between the expression of SERPINF1 and that of STAT1, CREM, and NR2F2 were calculated based on Spearman’s correlation analysis. The scatter plots displaying their correlations are presented in Figure 8.

## 4. Discussion

Glioma remains a therapeutic challenge to medical oncologists. Although the past decades have witnessed the advances in conventional radiochemotherapy and novel therapeutic interventions, the survival of glioma remains dismal, especially for GBM [27]. Hence, novel insights into the key molecular mechanisms underlying glioma progression and recurrence are crucial for precise treatment.

SERPINF1 is a multifunctional molecule with anti-angiogenic and neurotrophic activity [28]. It has also been reported that SERPINF1 is involved in various cancers. Notably, SERPINF1 seems to have bidirectional functions in different cancers. For example, SERPINF1 showed anti-tumor effects by inhibiting angiogenesis in melanoma, cholangiocarcinoma, and breast cancer [11,29,30]. However, many studies substantiated that SERPINF1 promotes the growth and metastasis of esophageal cancer, liver cancer, and ovarian cancer [12,13,31]. Li et al. reported that SERPINF1 exhibits contrasting intracellular and extracellular functions in hepatocellular carcinoma [14]. SERPINF1 could increase the intracellular accumulation of free fatty acids, thereby promoting the proliferation of hepatocellular carcinoma cells [14]. Meanwhile, extracellular SERPINF1 could suppress tumor angiogenesis in vivo, which explained why SERPINF1 did not affect the prognosis of patients with hepatocellular carcinoma [14].

The expression features and prognostic value of SERPINF1 in glioma have hitherto not been comprehensively documented in the literature. In the present study, we found that high expression of SERPINF1 correlated with risk factors of glioma, such as wild-type IDH, 1p19q non-codeletion, and a higher WHO grade. Furthermore, glioma patients with high SERPINF1 expression experienced poorer survival than those with low SERPINF1 expression. In addition, ROC curve analysis showed that SERPINF1 expression could predict the 3-, 5-, and 8-year survival rates of glioma patients with high AUC, suggesting SERPINF1 has huge prospects as a novel prognostic predictor for glioma.

Furthermore, the association between SERPINF1 expression and glioma stemness was investigated. GSCs are a group of tumor cells with limitless proliferation, infinite self-renewal, and multi-directional differentiation potentials, responsible for chemoradiotherapy resistance and tumor recurrence [32,33]. Bioinformatics analysis revealed that SERPINF1 expression was positively correlated with glioma stemness. Cellular experiments confirmed that SERPINF1 expression was significantly upregulated in GSCs. Interestingly, SERPINF1 has been substantiated to promote the self-renewal and multipotency of NSCs [19,34]. SERPINF1 could upregulate the expression of Notch downstream effectors such as HES1 and HES5, thereby antagonizing cell differentiation and promoting self-renewal [19,35]. In addition, SERPINF1 could also promote self-renewal and pluripotency maintenance in human embryonic stem cells by activating ERK1/2 signaling [36]. These results indicate that SERPINF1 may have a general positive effect on the self-renewal of stem cells. Moreover, the single-cell analysis showed that the Notch signaling pathway was substantially enriched in glioma cells with high SERPINF1 expression. It has been widely demonstrated that the Notch signaling pathway is specifically involved in preserving the self-renewal and survival of tumor stem cells [37,38]. Therefore, SERPINF1-related Notch signaling activation may contribute to stemness maintenance in glioma. The association between SERPINF1 upregulation and Notch signaling activation in glioma cells would be worthy of further investigation.

The effects of SERPINF1 knockdown on glioma cells and GSCs were also evaluated in vitro. Cellular functional experiments revealed that SERPINF1 knockdown significantly suppressed the proliferation, invasion, and migration of glioma cells A172 and LN18. Moreover, SERPINF1 knockdown impaired the sphere formation of GSC-A172 and GSC-LN18. The sphere formation assay has been widely used to evaluate stem cell activity [39,40,41]. In our study, we found that SERPINF1 expression was significantly upregulated when glioma cells were induced into GSCs. SERPINF1 silencing may perturb the transformation of glioma cells into GSCs, indicating that SERPINF1 is a potential target for glioma stemness.

Given the essential role of SERPINF1 in glioma, we performed SCENIC analysis to infer the potential TFs that may modulate the expression of SERPINF1. The results showed that STAT1, CREM, and NR2F2 were the most likely upstream TFs of SERPINF1 in glioma. Interestingly, previous studies have reported the essential roles of STAT1 and NR2F2 in the malignancy and stemness of tumors. For example, KAOWINN et al. reported that STAT1 silencing decreased the expression of stemness-related genes, thereby inhibiting sphere formation of lung cancer stem cells [42]. Chen et al. corroborated that STAT1 upregulation promoted the malignant behaviors of colorectal cancer stem cells [43]. In addition, Mauri et al. revealed that NR2F2 could promote the stemness of squamous cell carcinoma [44]. Qin et al. reported that NR2F2 promoted prostate tumorigenesis by inhibiting the TGF-β-induced growth barrier [45]. However, the role of CREM in tumors has not been thoroughly investigated. Importantly, the prediction of TFs may improve our understanding of the mechanisms underlying the transcriptional upregulation of SERPINF1 in glioma.

## 5. Conclusions

High SERPINF1 expression portends a poor prognosis in glioma patients. In vitro experiments confirmed that SERPINF1 knockdown could suppress the proliferation, invasion, and migration of glioma cells. In addition, SERPINF1 is expected to be a stem cell marker in glioma. The expression of SERPINF1 is significantly upregulated in GSCs, and SERPINF1 knockdown impairs the sphere formation of GSC-A172 and GSC-LN18, indicating that SERPINF1 is a potential target for glioma stemness. Single-cell analysis showed that Notch signaling activation was closely associated with high SERPINF1 expression, warranting further investigation. Furthermore, transcriptional factors STAT1, CREM, and NR2F2 may contribute to the high expression of SERPINF1. In a nutshell, our study uncovered the prognostic significance and molecular functions of SERPINF1, suggesting that SERPINF1 is a candidate therapeutic target for glioma.

## Figures and Tables

**Figure 1 genes-14-00580-f001:**
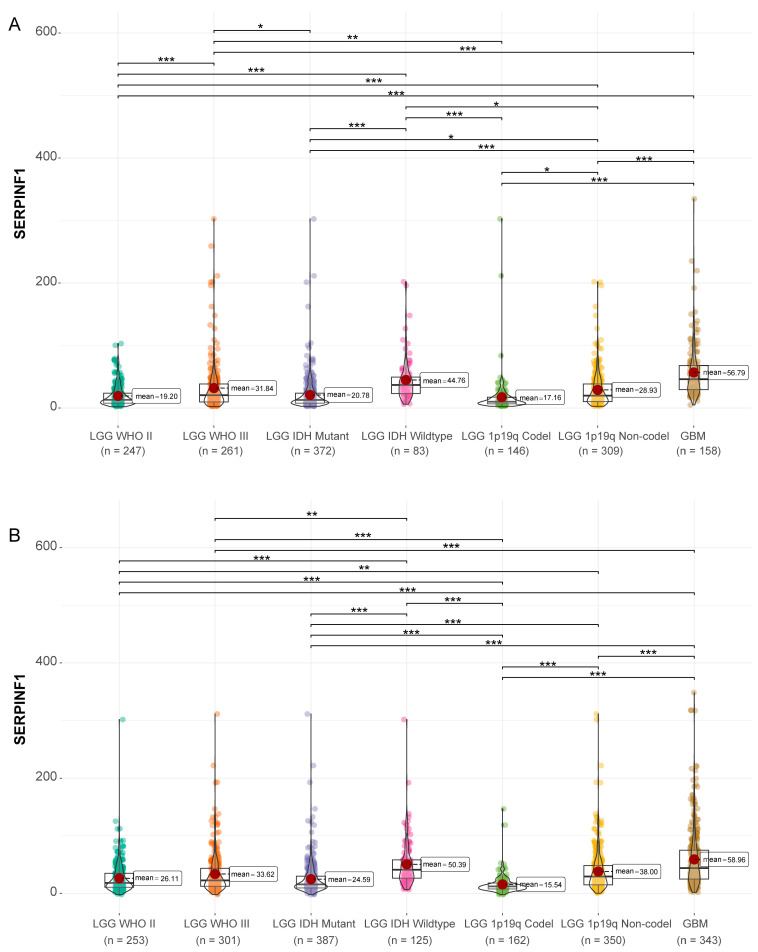
The expression of SERPINF1 in gliomas of different grades and molecular subtypes in TCGA (**A**) and CGGA (**B**). (* *p* < 0.05, ** *p* < 0.01, *** *p* < 0.001).

**Figure 2 genes-14-00580-f002:**
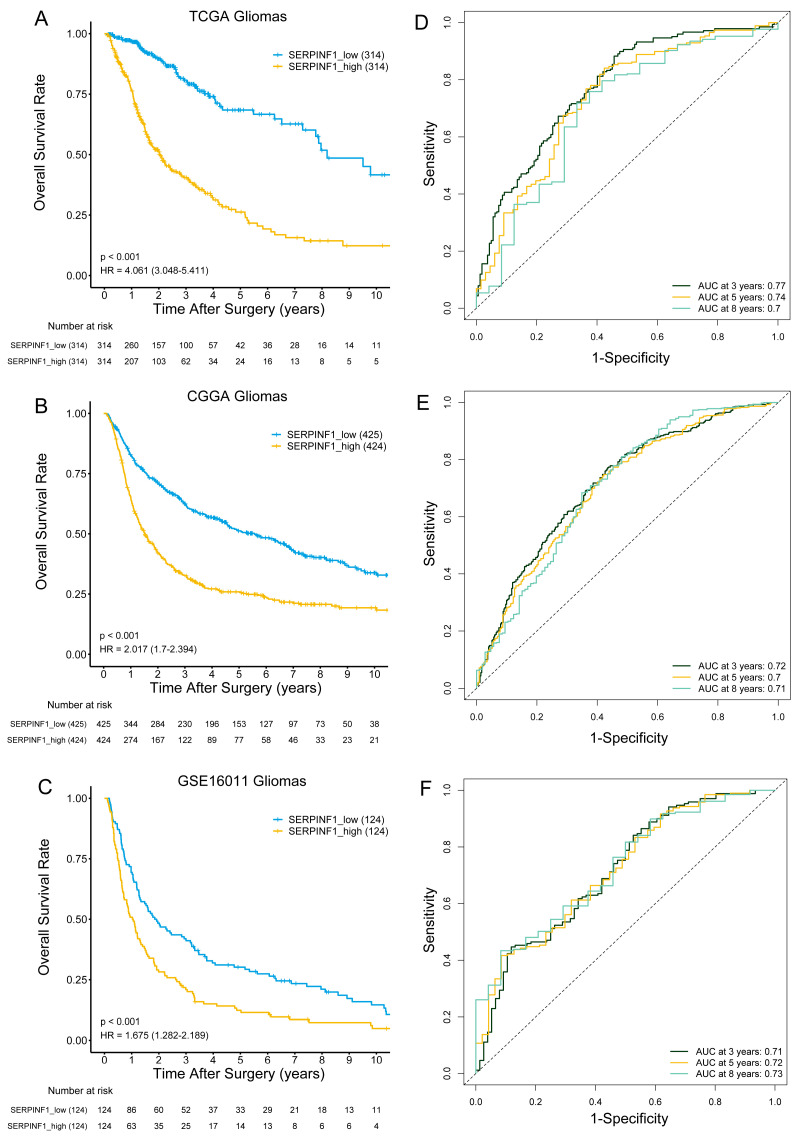
The prognostic value of SERPINF1 expression in glioma. Kaplan–Meier survival curves of the high- and low-SERPINF1 groups in TCGA (**A**), CGGA (**B**), and GSE16011 (**C**). Time-dependent ROC curves for predicting 3-, 5-, and 8-year survival based on SERPINF1 expression in TCGA (**D**), CGGA (**E**), and GSE16011 (**F**).

**Figure 3 genes-14-00580-f003:**
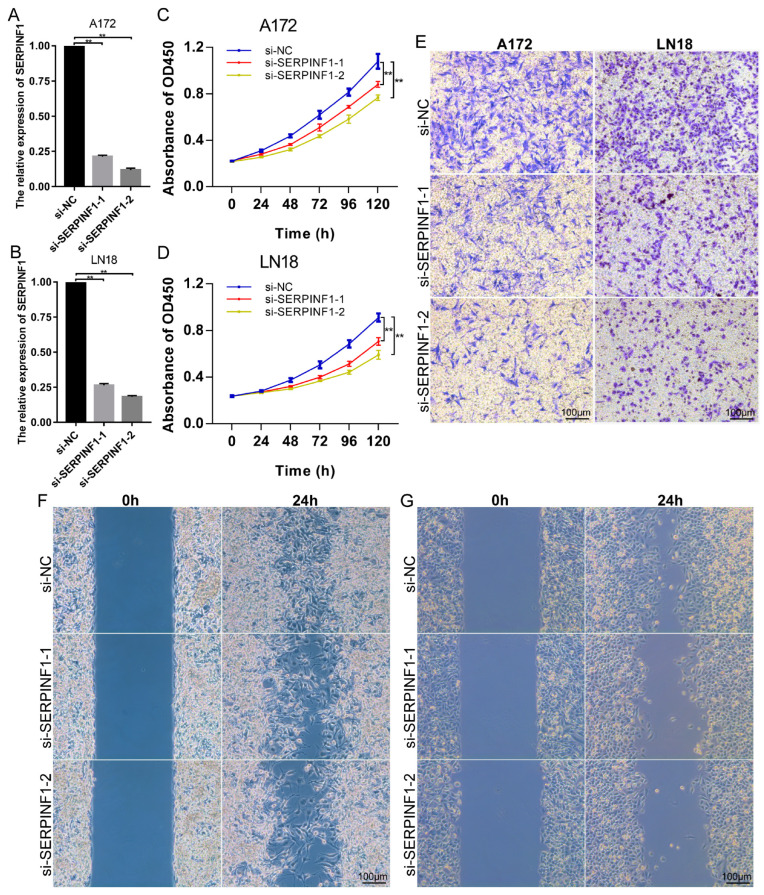
Effects of SERPINF1 knockdown on the proliferation and invasion of glioma cells. (**A**,**B**) Knockdown efficiency of SERPINF1 in A172 and LN18 verified via RT-qPCR. (**C**,**D**) Anti-proliferative effect of SERPINF1 knockdown on A172 and LN18 verified via CCK-8 assay. (**E**) Anti-invasive effect of SERPINF1 knockdown on A172 and LN18 verified via Transwell assay. (**F**,**G**) Anti-migrative effect of SERPINF1 knockdown on A172 (**F**) and LN18 (**G**) verified via migration assay. (** *p* < 0.01).

**Figure 4 genes-14-00580-f004:**
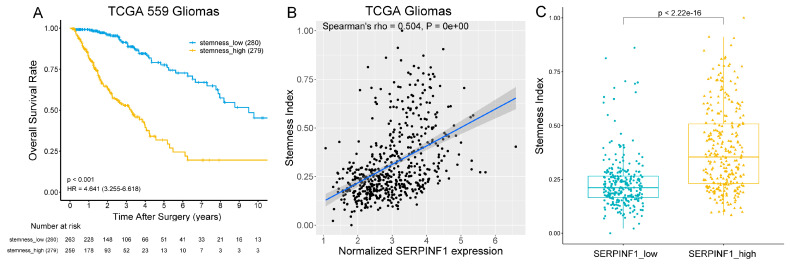
The association between SERPINF1 expression and glioma stemness. (**A**) Kaplan–Meier survival curves based on glioma stemness in TCGA. (**B**) Spearman’s correlation analysis between SERPINF1 expression and glioma stemness. (**C**) The difference in glioma stemness between high- and low-SERPINF1 groups.

**Figure 5 genes-14-00580-f005:**
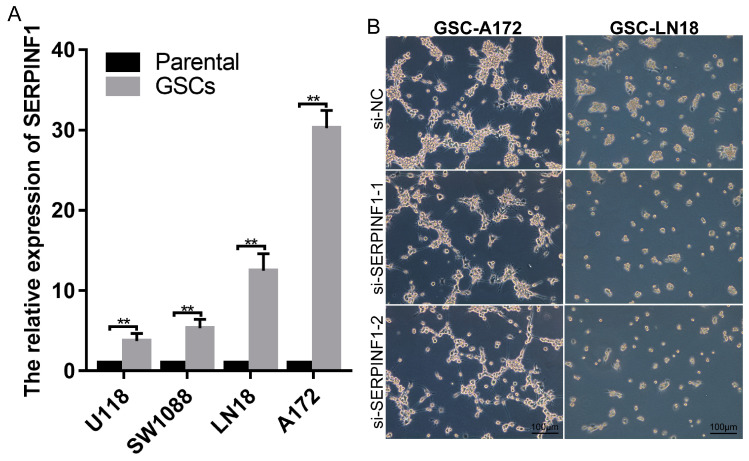
The role of SERPINF1 in GSCs. (**A**) Difference in SERPINF1 expression between GSCs and their parental cell lines. (**B**) The effect of SERPINF1 knockdown on the sphere formation of GSC-A172 and GSC-LN18. (** *p* < 0.01).

**Figure 6 genes-14-00580-f006:**
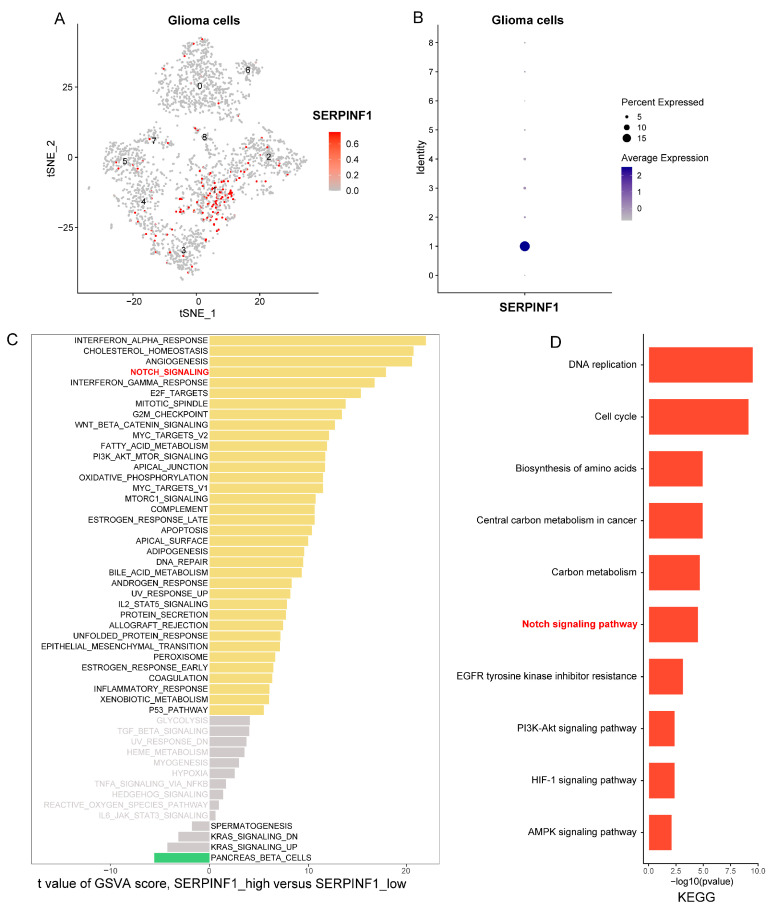
The signaling pathways associated with SERPINF1 expression at the single-cell level. (**A**) tSNE plot of the 2751 glioma cells with each cell color-coded based on the expression of SERPINF1. (**B**) SERPINF1 expression in nine subclusters of glioma cells visualized in a bubble plot. (**C**) Differences in pathway activities scored per cell via GSVA between high- and low-SERPINF1 clusters. (**D**) Signaling pathways enriched in high-SERPINF1 cluster based on KEGG analysis at the single-cell level.

**Figure 7 genes-14-00580-f007:**
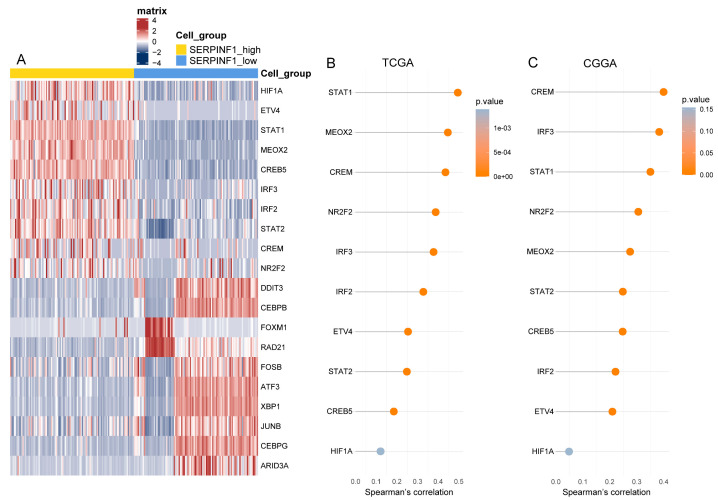
Transcriptional regulation of SERPINF1 in glioma. (**A**) Heatmap of TF activity via SCENIC at the single-cell level. The correlations between the expression of TFs and that of SERPINF1 verified via RNA-seq data of gliomas in TCGA (**B**) and CGGA (**C**).

**Figure 8 genes-14-00580-f008:**
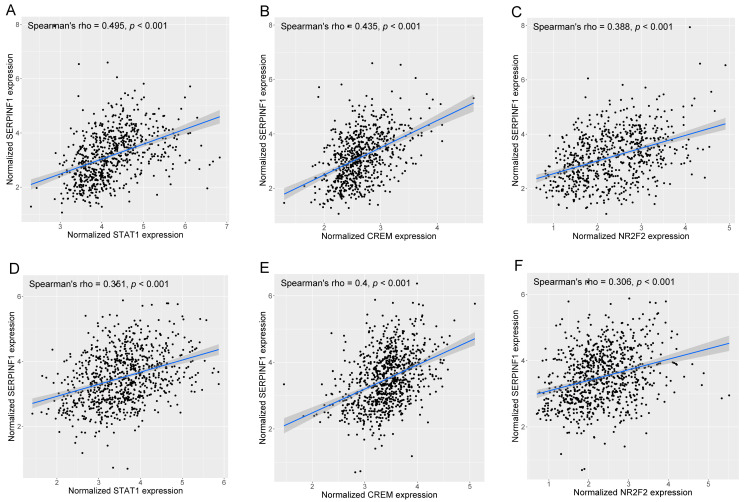
The correlations between the expression of SERPINF1 and that of STAT1, CREM, and NR2F2 in TCGA (**A**–**C**) and CGGA (**D**–**F**).

## Data Availability

The data analyzed in this study can be found in online repositories. The names of the repositories and accession numbers are included in the article.

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
