# Peer review of "SERPINF1 Mediates Tumor Progression and Stemness in Glioma"

_genes, 2023, doi:10.3390/genes14030580_

Round 1
Reviewer 1 Report
· Overall, the paper was well-written and somehow clear.
· But there are several grammatical errors, and some confusing sentences need to rectify. Suggest the manuscript should have been read and edited by a native English speaker with knowledge of the techniques.
· Abstract: Use italic for “in vitro” or add dash symbol (in-vitro)
· Abstract need to be revised à be more specific by including some results as current is more to general statements.
· Section 2.8 à need detail explanation of methodology.
· Need to include scale bar in all microscopic images = Figure 3 and 5
· Line 381 – 282 : “SERPINF1-related Notch signaling activation may contribute to the stemness maintenance in glioma” à need further and detail discussion on how SERPINF1 regulate Notch signaling
· Further discussion on “SERPINF1 knockdown impaired the sphere formation of GSC-A172 and GSC-LN18” at Line 386, is needed. How can the knockdown reduced sphere formation? Any specific pathways or proteins will cause this phenomenon?
· Conclusion is too simple - need to revise and highlight the main outcomes of the study.
Author Response
Response to Reviewer 1 Comments
Comments and Suggestions for Authors
Overall, the paper was well-written and somehow clear.
Dear reviewer,
We are so grateful for your critical evaluation on our manuscript. Based on your suggestions, we have answered the questions point-by-point in detail. All the changes are marked in the revised manuscript. The main corrections and responses to your comments are listed below.
- But there are several grammatical errors, and some confusing sentences need to rectify. Suggest the manuscript should have been read and edited by a native English speaker with knowledge of the techniques.
Response 1: Thank you for your suggestion. Our manuscript has been edited by a native English speaker.
- Abstract: Use italic for “in vitro” or add dash symbol (in-vitro)
Response 2: Thank you for your suggestion. We have made the correction. (revised manuscript, line 12)
- Abstract need to be revised à be more specific by including some results as current is more to general statements.
Response 3: Thank you for your suggestion. We have added some specific results in abstract.
- Section 2.8 à need detail explanation of methodology.
Response 4: Thank you for your suggestion. We have supplemented the methodology in Section 2.8. (revised manuscript, lines 135-140)
- Need to include scale bar in all microscopic images = Figure 3 and 5
Response 5: Thank you for your suggestion. We have added scale bars in all microscopic images.
- Line 381 – 282 : “SERPINF1-related Notch signaling activation may contribute to the stemness maintenance in glioma” à need further and detail discussion on how SERPINF1 regulate Notch signaling
Response 6: Thank you for your suggestion. We have added a further discussion on the role of SERPINF1 in Notch signaling. (revised manuscript, lines 452-464)
- Further discussion on “SERPINF1 knockdown impaired the sphere formation of GSC-A172 and GSC-LN18” at Line 386, is needed. How can the knockdown reduced sphere formation? Any specific pathways or proteins will cause this phenomenon?
Response 7: Thank you for your suggestion. Sphere formation assay has been widely used to evaluate stem cell activity [1-3]. In our study, we found that SERPINF1 expression was significantly upregulated when glioma cells were induced into GSCs (Figure 5A). SERPINF1 silencing may perturb the transformation of glioma cells into GSCs, thereby reducing sphere formation. Stem cell activity is regulated by multiple signaling pathways. It is difficult for us to explore and validate the stem cell-related signaling pathways regulated by SERPINF1 this time. In the follow-up work, we will further study the molecular mechanism of SERPINF1 in glioma in vitro and in vivo. By that time, we will try to find out and validate the stem cell-related signaling pathways regulated by SERPINF1. We respectfully request your understanding and tolerance. (revised manuscript, lines 469-472)
- Conclusion is too simple - need to revise and highlight the main outcomes of the study.
Response 8: Thank you for your suggestion. We have supplemented the main results in Conclusion.
Again, we appreciate your insightful comments. Thank you for taking the time to help us improve the paper.
References
- J. Manuel Iglesias, I. Beloqui, F. Garcia-Garcia, O. Leis, A. Vazquez-Martin, A. Eguiara, et al., Mammosphere formation in breast carcinoma cell lines depends upon expression of E-cadherin, PLoS One. 8(10) (2013) e77281.
- Y. Shi, O.A. Guryanova, W. Zhou, C. Liu, Z. Huang, X. Fang, et al., Ibrutinib inactivates BMX-STAT3 in glioma stem cells to impair malignant growth and radioresistance, Sci. Transl. Med. 10(443) (2018).
- D. Yue, Z. Zhang, J. Li, X. Chen, Y. Ping, S. Liu, et al., Transforming growth factor-beta1 promotes the migration and invasion of sphere-forming stem-like cell subpopulations in esophageal cancer, Exp. Cell Res. 336(1) (2015) 141-9.

Reviewer 2 Report
Comments
Manuscript title: SERPINF1 mediates tumor progression and stemness in glioma
Authors: Song et al.
Manuscript ID: genes-2212042
Novel gene, SERPINF1, has been implicated in many cancers including glioma. Many investigators have demonstrated that overexpression of SERPINF1 both at the mRNA and protein levels in cancer as compared with normal tissues and has been significantly associated with poor prognosis.
Major concerns:
The authors have extensively analyzed publicly available gene expression data from thousands of studies to demonstrate that SERPINF1 serves as a promising predictor for clinical diagnosis, prognosis, and therapeutic response of glioma patients, providing a novel and reliable target for glioma treatment. However, the biological validation of the bioinformatic data from this study is minimal.
The authors need to address the following:
1. In addition to Real-time mRNA expression after silencing and cell proliferation analysis (Fig. 3A and B), the authors need to assess the protein expression level.
2. The authors should include SERPINF1 expression level both in the glioma cell lines and normal human astrocytes by Real-time and Western blot analysis.
3. Because SERPINF1 associated with Notch signaling, E2F targets, mitotic spindle, G2M checkpoint, wnt/β-266 catenin signaling, and MYC targets the authors need to show
a. Cellular, nuclear and morphological changes after genetic silencing of SERPINF1.
b. induction of cellular autophagy and inhibition of migration after genetic silencing.
4. If there is any known pharmacological inhibitor available for SERPINF1, the authors need to examine the effect on cell proliferation inhibition and induction of apoptosis.
Minor concerns:
Methods:
Quantitative real-time polymerase chain reaction (qRT-PCR) analysis: the authors need to include the appropriate reference to show how they calculated 2−ΔΔCt.
Author Response
Response to Reviewer 2 Comments
Comments and Suggestions for Authors
Comments
Manuscript title: SERPINF1 mediates tumor progression and stemness in glioma
Authors: Song et al.
Manuscript ID: genes-2212042
Novel gene, SERPINF1, has been implicated in many cancers including glioma. Many investigators have demonstrated that overexpression of SERPINF1 both at the mRNA and protein levels in cancer as compared with normal tissues and has been significantly associated with poor prognosis.
Dear reviewer,
We are so grateful for your critical evaluation on our manuscript. Based on your suggestions, we have answered the questions point-by-point in detail. All the changes are marked in the revised manuscript. The main corrections and responses to your comments are listed below.
Major concerns:
The authors have extensively analyzed publicly available gene expression data from thousands of studies to demonstrate that SERPINF1 serves as a promising predictor for clinical diagnosis, prognosis, and therapeutic response of glioma patients, providing a novel and reliable target for glioma treatment. However, the biological validation of the bioinformatic data from this study is minimal.
The authors need to address the following:
- In addition to Real-time mRNA expression after silencing and cell proliferation analysis (Fig. 3A and B), the authors need to assess the protein expression level.
Response 1: Thank you for your suggestion. We have confirmed that the expression of SERPINF1 was decreased on the protein level. (Supplementary Figure S1)
- The authors should include SERPINF1 expression level both in the glioma cell lines and normal human astrocytes by Real-time and Western blot analysis.
Response 2: Thank you for your suggestion. Regrettably, due to the difficulty in obtaining normal human astrocytes, it is hard for us to supplement this experiment this time. Thank you for your understanding.
- Because SERPINF1 associated with Notch signaling, E2F targets, mitotic spindle, G2M checkpoint, wnt/β-266 catenin signaling, and MYC targets the authors need to show
- Cellular, nuclear and morphological changes after genetic silencing of SERPINF1.
Response 3a: Thank you for your suggestion. We have supplemented the optical microscope images of A172 and LN18 cells. However, due to the lack of an electron microscopy, we are unable to observe the ultrastructure of cytoplasm and nucleus. Thank you for your understanding. (Supplementary Figure S2)
- induction of cellular autophagy and inhibition of migration after genetic silencing.
Response 3b: Thank you for your suggestion. Cellular autophagy verification is a big project, involving transmission electron microscopy scanning and detection of complex signal pathways. We have very little experience in this area. It is hard for us to supplement the cellular autophagy detection this time. In the follow-up work, we will further study the molecular mechanism of SERPINF1 in glioma in vitro and in vivo. By that time, we will try to detect the cellular autophagy after SERPINF1 silencing. We respectfully request your understanding and tolerance. In addition, we have supplemented the cell migration assay according to your advice. (Figure 3F, G)
- If there is any known pharmacological inhibitor available for SERPINF1, the authors need to examine the effect on cell proliferation inhibition and induction of apoptosis.
Response 4: Currently, there is no pharmacological inhibitor available for SERPINF1. Thank you for your advice.
Minor concerns:
Methods:
Quantitative real-time polymerase chain reaction (qRT-PCR) analysis: the authors need to include the appropriate reference to show how they calculated 2−ΔΔCt.
Response 5: Thank you for your suggestion. We have made the recommended change. (revised manuscript, line 117)
Again, we appreciate your insightful comments. Thank you for taking the time to help us improve the paper.

Round 2
Reviewer 2 Report
The manuscript would be much improved should the authors made some effort to address all the comments.